# OpenReview forum: "Control Tax: The Price of Keeping AI in Check"
_ICLR.cc/2026/Conference — ICLR 2026 Poster_

### Official Review · Reviewer_S345 · 2025-10-15

**Soundness:** 3
**Presentation:** 1
**Contribution:** 3
**Rating:** 6
**Confidence:** 5

**Summary:**

This paper introduces the concept of Control Tax—the operational and financial cost of implementing oversight mechanisms in AI systems, especially under the framework of AI Control (AIC). It develops a quantitative framework linking model monitoring performance with long-term system safety and cost-efficiency. The authors simulate adversarial conditions using code backdooring tasks in the APPS dataset, where attacker LMs introduce subtle flaws, and monitor LMs attempt to detect them. Through empirical ROC analyses and game-theoretic modeling, the study quantifies how safety scales with auditing budgets, model performance, and financial cost.

The paper’s main contribution is establishing a principled economic-safety trade-off in AI oversight. It demonstrates that stronger monitors do not always yield proportionally higher safety—cost-effective combinations of models and auditing strategies can outperform expensive monitors under limited budgets. The theoretical results extend to batch and sequential deployment scenarios, offering practical guidelines: conceal deployment horizons, audit heavily early on, and balance cost with monitoring quality. Ultimately, the work provides a foundational step in formalizing the economics of AI safety, giving researchers and practitioners a concrete basis for optimizing control strategies in real-world deployments.

**Strengths:**

The paper presents an original and timely concept—Control Tax—that quantifies the trade-off between AI oversight cost and safety, linking economic modeling with AI control in a novel way. The methodology is solid, combining theoretical analysis with well-designed experiments on code-based adversarial tasks. The writing is clear and effectively supported by figures illustrating key trade-offs. Overall, the paper makes a significant contribution by formalizing the economics of AI safety and offering practical insights for cost-efficient oversight in real-world AI systems.

**Weaknesses:**

The paper’s evaluation is limited to synthetic and code-based scenarios, which may not fully capture the complexity of real-world AI oversight challenges. Some assumptions in the theoretical modeling—such as simplified attacker–monitor dynamics and fixed audit budgetsreduce the generality of its conclusions. The framework’s practical applicability could be better demonstrated through broader experiments or deployment case studies. Additionally, parts of the mathematical exposition are dense, which may hinder accessibility for readers less familiar with economic modeling in AI safety.

**Questions:**

How sensitive are the results to the choice of monitoring and attack models? Would performance trends hold under different model architectures or non-code tasks?

Could the authors provide more empirical validation on real-world or human-in-the-loop oversight settings to strengthen practical relevance?

Theoretical results assume fixed budgets and known audit horizons—how would the Control Tax framework adapt to dynamic or uncertain resource environments?

Have the authors explored nonlinear or diminishing returns effects in monitoring performance as spending increases, and how might that influence policy implications?

Can the authors elaborate on how Control Tax could integrate with governance mechanisms such as red-teaming or model certification frameworks to enhance real-world applicability?

---

> ### Author Response · Authors · 2025-11-26
> **Response to Reviewer S345 (1/2)**
>
> We are grateful for the Reviewer’s assessment that the Control Tax framework is “timely and original,” that our methodology is “solid,” and thats the paper constitutes a “significant contribution.” Below, we respond to the limitations raised.
>
> > The paper’s evaluation is limited to synthetic and code-based scenarios, which may not fully capture the complexity of real-world AI oversight challenges.
>
> We agree with the reviewer that broader evaluations of AI control on non-code-related tasks would be beneficial. Since this was a shared concern among the reviewers, we have now added an evaluation on SHADE-Arena [2] (a non-code domain with long horizon agentic interactions). We explain the setup in more detail and provide analysis of the results in the response to Reviewer 97WH.
>
> > Some assumptions in the theoretical modeling—such as simplified attacker–monitor dynamics and fixed audit budgets reduce the generality of its conclusions.
>
> We agree that the changing budgets might be a factor influencing deployment of AI control. In that situation, what we call “anytime sequential trusted monitoring” might be a better model for long-term deployments. We provide the definition, a preliminary discussion, and an experimental evalution in Appendix G. Depending on the attacker’s $p_+$ and the monitor ROC curve, the optimal strategy for the attacker might either be to wait indefinitely as the monitoring budget shrinks, or to attack with a finite fixed probability.
>
> > The framework’s practical applicability could be better demonstrated through broader experiments or deployment case studies.
>
> We agree with the Reviewer and have therefore conducted an additional experiment on the SHADE-Arena framework [2]. We provide the details and discussion in our response to Reviewer 97WH. The overall conclusion that a minor gain in safety at the highest level comes with a drastic increase in costs does generalize to this setting as well.
>
> > Additionally, parts of the mathematical exposition are dense, which may hinder accessibility for readers less familiar with economic modeling in AI safety.
>
> We tried to be conscious of the varying familiarity of the readers with the mathematical modeling notions that we used. To that end, in Section 4 we provide a discussion with the high-level takeaways from the theory in Section 3. We have highlighted these takeaways more explicitly in the new revision of the paper.
>
> ## Questions
> > How sensitive are the results to the choice of monitoring and attack models? Would performance trends hold under different model architectures or non-code tasks?
>
> We are motivated to provide a comprehensive overview of the models’ capabilities in the paper, with 12 different attacker and 13 monitor models in total. Our results in Figure 3 broadly conform with the general performance trends, with more expensive and newer models performing better as both attackers and monitors. While the increase in monitor performance is smooth, the attacker performance has “plateaus”: the newer model families like GPT-4.1 or Gemini 2.5 mark a step improvement in subtlety of attacks. We also show which fraction $p_+$ of the backdoors is successful for a subset of the attackers in Figure 9 in the Appendix. This probability also shows predictable increases as models get more capable.
>
> With regard to varying the architecture, it would be hard to tell because we experiment with strong proprietary models and do not have access to the model architectures. We assume that through an evaluation with a large set of models from three different providers we capture variations in the architectures of the frontier language models.
>
> Our new experiments with the SHADE-Arena environment, which we discuss in more detail in the response to Reviewer 97WH, also show that the broad trends that we outlined remain in more complex and non-code domains. Importantly, a small extra improvement in safety (from ~0.73 to ~0.78) requires a large step change in the monitoring costs (from USD 20 to USD 120 per 1000 samples).

---

> > ### Author Response · Authors · 2025-11-26
> > **Response to Reviewer S345 (2/2)**
> >
> > > Could the authors provide more empirical validation on real-world or human-in-the-loop oversight settings to strengthen practical relevance?
> >
> > Yes, we provide an evaluation on the SHADE-Arena framework from [2] in our response to Reviewer 97WH. This framework is composed of more realistic administrative tasks, with each task requiring many agentic interactions with the environment.
> >
> > > Theoretical results assume fixed budgets and known audit horizons—how would the Control Tax framework adapt to dynamic or uncertain resource environments?
> >
> > We provide a preliminary discussion of a more general sequential anytime monitoring setting in Appendix G, alongside an experimental evaluation. This setting can be used to model a changing auditing budget. The main change in the modeling there is that we need to be more mindful of the optimal attacker strategy, as it is sometimes beneficial for the attacker to be maximally patient and attack extremely rarely. We also discuss this phenomenon in Section 4.
> >
> > For a dynamic environment, the two main variables to track would be the attacker’s plausible sabotage success probability $p_+$ and the monitor’s ROC curves. In such situations, continuous red-teaming during deployment is warranted. For a discussion of how AI control measures can be deployed in practice under variable and uncertain conditions, we kindly refer the Reviewer to [3,4].
> >
> > > Have the authors explored nonlinear or diminishing returns effects in monitoring performance as spending increases, and how might that influence policy implications?
> >
> > Indeed, we discuss the effect of diminishing returns on investment w.r.t. the safety of the protocol in Section 6. One policy implication here is that one can deploy a control protocol with most of the safety guarantees rather cheaply, warranting broader deployment. For researchers, this means that focus on cheaper safety measures might be more beneficial than focus on significantly more expensive but marginally more comprehensive measures. Another implication is that the human auditing budget $c$ can and should be explicitly traded off against deployment of more expensive monitors.
> >
> > > Can the authors elaborate on how Control Tax could integrate with governance mechanisms such as red-teaming or model certification frameworks to enhance real-world applicability?
> >
> > High-stakes AI control is most relevant for internal deployments in the frontier AI labs. Publicly available AI agents can generally be governed more easily through pre-determined safety thresholds. This makes the tax framing less relevant, since the deployer can be forced to bear the costs associated with hitting the safety threshold. At the same time, even for public regulation, control tax evaluations can play a role for the policymakers when deciding on reasonable thresholds. The framework of low-stakes control [5] might also be better suited for broader, external deployments.
> >
> > Internal deployments, on the other hand, are inherently harder to get an insight into, because companies have legitimate reasons to keep their internal operations non-transparent. The Control Tax framing is thus more applicable, since cheaper measures make it more appealing for the company to choose to deploy control protocols under tight budget constraints.
> >
> > While we believe that there is a strong case for control tax being relevant for policy decisions, we acknowledge that we are not governance experts, and thus have to leave the specific implementation of governance measures to policymakers.
> >
> > We hope this addresses your concerns, and we are happy to answer any further questions you have!
> >
> > -----
> > [1] Kutasov, Jon, et al. "Evaluating Control Protocols for Untrusted AI Agents." arXiv preprint arXiv:2511.02997 (2025).
> >
> > [2] Kutasov, Jonathan, et al. "SHADE-Arena: Evaluating Sabotage and Monitoring in LLM Agents." arXiv preprint arXiv:2506.15740 (2025).
> >
> > [3] Korbak, Tomek, et al. "A sketch of an ai control safety case." arXiv preprint arXiv:2501.17315 (2025).
> >
> > [4] Korbak, Tomek, et al. "How to evaluate control measures for LLM agents? A trajectory from today to superintelligence." arXiv preprint arXiv:2504.05259 (2025).
> >
> > [5] Wen, Jiaxin, et al. "Adaptive deployment of untrusted llms reduces distributed threats." arXiv preprint arXiv:2411.17693 (2024).

---

> > > ### Comment · Reviewer_S345 · 2025-11-26
> > > **Understood. Improved ratings**
> > >
> > > Understood. Improved ratings

---

### Official Review · Reviewer_WjCo · 2025-10-31

**Soundness:** 3
**Presentation:** 3
**Contribution:** 3
**Rating:** 6
**Confidence:** 2

**Summary:**

The manuscript introduces the concept of a Control Tax, defined as the operational and financial cost required to maintain effective oversight in AI deployment pipelines. In the presented setting, a potentially misaligned attacker model inserts subtle backdoors into code solutions from the APPS dataset, while a monitor model attempts to detect these vulnerabilities. The manuscript develops a two-part contribution: first, it empirically evaluates a wide range of modern language models as both attackers and monitors to estimate their detection and evasion capabilities; second, it provides a game-theoretic framework that translates a monitor’s ROC curve and auditing budget into a quantifiable probability of avoiding compromise during long-term deployment. The resulting Control Tax curves illustrate the trade-off between safety and auditing cost and show how optimal monitoring strategies can differ depending on attacker capability, resource constraints, and whether the deployment is batch or sequential.

**Strengths:**

The manuscript’s core contribution is the introduction of the Control Tax as a variable linking monitoring system performance to operational and financial deployment costs. This formulation is original in that it treats oversight not only as a technical alignment problem but as a resource allocation problem. Prior work in AI control has largely focused on proving that monitoring protocols can succeed in principle; this manuscript instead addresses the conditions under which they are practically adoptable. The resulting framework allows different safety targets to be expressed in terms of budget and auditing strategy, which provides a concrete decision structure for practitioners considering real deployment contexts.

The empirical component is constructed with clear methodological discipline. The attacker and monitor models are evaluated using a standardized APPS coding task environment, which ensures controlled variation across experiments. Backdoored solutions are generated using an agentic multi-step pipeline that enforces correctness on public test cases and explicitly validates malicious triggers, ensuring that the resulting dataset meaningfully distinguishes intentional vulnerabilities from ordinary programming errors. The monitoring task is operationalized through a calibrated suspiciousness scoring procedure, rather than binary detection, which makes the measurements directly compatible with downstream auditing-budget analyses. The breadth of models evaluated, combined with balanced datasets and formatting normalization (e.g., comment stripping), helps reduce confounds arising from stylistic artifacts.

The theoretical modeling is presented with clarity. The manuscript explains how ROC curves derived from the monitoring evaluations map into dynamic auditing decisions in both batch and sequential settings. The derivation of safety probabilities under optimal attacker and defender strategies is shown with reference to established stochastic process frameworks, and intermediate assumptions are made explicit. The paper consistently connects the modeling back to operational interpretations, explaining not just the mathematical structure but also how the defender’s auditing schedule should be shaped across time.

The significance of the work lies in providing a tractable bridge between monitor accuracy metrics and deployment-level safety guarantees. Organizations deciding whether to deploy model oversight systems require not only performance comparisons but also cost and scheduling implications. This manuscript offers a structured method for inferring how much auditing is needed to maintain a desired safety level and how that cost changes when adopting a stronger or weaker monitor. These results directly inform strategy discussions in AI governance, red-teaming design, and scalable oversight implementation.

**Weaknesses:**

The primary limitation of the work lies in the assumption of a stateless, non-adaptive attacker. In the empirical pipeline, the attacker generates backdoors independently and does not update strategy based on audit outcomes or monitor behavior. In practice, attackers (human or model-based) may iteratively probe the monitoring boundary, learn suspiciousness patterns, and reduce detection rates over time. The theoretical analysis also assumes that the red team maintains a fixed insertion probability optimized only with respect to the task horizon. Extending the framework to adaptive attackers—for example, those modeling the monitor as a learnable discriminator—would materially strengthen claims about real-world robustness. The manuscript acknowledges this scenario but does not examine parameter regimes in which adaptivity meaningfully alters the Control Tax.

The evaluation domain is restricted to code backdooring in APPS. This setting has several desirable properties (e.g., clear correctness signals, deterministic vulnerability definitions), but it differs from domains where oversight is often considered most critical, such as multi-step planning, agentic task execution, or socially mediated harm. The specific structure of programming tasks—particularly the existence of public test cases that constrain solution space—may limit the diversity of viable backdoors and thus the challenge imposed on monitors. It remains uncertain whether the same monitoring strategies and cost-safety trade-offs apply when failures are less crisply defined or when correctness requires implicit world modeling. A discussion of transferability, or a small-scale extension to one additional domain with less explicit ground truth, would clarify the scope of applicability.

The Control Tax curves depend on model cost estimates treated as fixed. In real deployment settings, inference cost varies substantially across hardware configurations, batching strategies, and licensing models. While this does not undermine the conceptual framework, it means the curves are best interpreted as relative guidance rather than directly actionable cost predictions. The manuscript could strengthen its practical significance by characterizing sensitivity to cost assumptions (e.g., showing how safety frontiers change when monitor cost decreases by an order of magnitude, or when attackers and monitors use locally deployed models instead of hosted APIs).

**Questions:**

How does the framework change if the attacker adapts to the monitor’s observed actions over time?

How sensitive are the cost-benefit conclusions to different pricing assumptions for inference compute?

---

> ### Author Response · Authors · 2025-11-26
> **Response to Reviewer WjCo**
>
> We appreciate that the Reviewer finds “clear methodological discipline” in our experimental setup, finds our theoretical sections to be “presented with clarity,” and acknowledges the relevance of our framework to strategical discussions around AI governance and deployment. Below we address the main concerns raised by the Reviewer and answer questions
> ## Statelessness assumption
> We thank the Reviewer for pointing out the attacker statelessness in AI control. This assumption, first introduced by Greenblatt et al [1], is a foundational assumption in AI control. Following [1] (footnote 4), we justify it by the fact that the blue team has control over what goes into the context / training data of the untrusted model. In the control framework, attacker state is not preserved between independent tasks since the blue team can clear the untrusted model’s context once the new task begins. Therefore, the strategy where the attacker “may iteratively probe the monitoring boundary,” suggested by the reviewer, cannot be used efficiently since the attacker will not retain the context of past actions. We would also like to note that the attacker does not have access to the monitor scores.
>
> For long-horizon agentic deployments, the attacker can infer some information about the monitor if it can discriminate which actions in the history have been deferred. Since the APPS setting is single-step, this strategy is not viable for it. One can also imagine that an advanced attacker can leave hidden messages to itself e.g. via its interaction with the web to retain some information about the monitor and the past attack attempts. However, this capability would require the attacker to communicate with itself steganographically, without triggering the monitoring system. Steganographic communcation capabilities in AI are an ongoing area of research [2,3], and we consider it to be out of scope for this work.
> ## Limited evaluation
> We agree with the reviewer that another evaluation domain would strengthen the contribution of the paper. We therefore ran an extra evaluation on the SHADE Arena evaluation framework from Kutasov et al [4]. We kindly refer to our response to Reviewer 97WH for the results and discussion.
>
> ## Sensitivity to cost
> The exact control tax curves from our paper indeed change with the costs of API access. A cost-cosnscious user may also self-host the monitor, in which case the cost becomes dependent on the exact hardware available. The primary use of the high-stakes control framework is for internal deployments within the labs (see Section 2 in [1]), and we cannot know the actual costs of running the protocols internally. Using the API costs is a noisy approximation, but, as we show in the paper, this can already provide useful insights.
>
> Overall, our main contribution in the paper is the conceptual framework rather then the exact cost estimates. We agree with the reviewer that the framework is not undermined by variations in the cost. If the practitioners choose to adopt this framework for internal control evaluations, they would have a better understanding of the costs involved in monitoring for their scenarios.
>
> -----
>
> ## Questions
> > How does the framework change if the attacker adapts to the monitor’s observed actions over time?
>
> As we discuss above, lack of state preservation between tasks is foundational to control due to blue team having access to the untrusted model’s context. If it we to be reintroduced, the framework would more closely resemble traditional security practices such as security from insider threats (where insiders are humans and therefore do maintain state), see e.g. [5] for a qualitative comparison.
>
> > How sensitive are the cost-benefit conclusions to different pricing assumptions for inference compute?
>
> The cost-benefit analysis changes significantly for internal deployments within AI labs, since the labs likely have access to their own models for much cheaper. Open-source self-hosted models can also change the relative costs picture. However, the phenomenon of cheap models being a preferable choice even with worse performance is likely to remain in place due to the plurality of the available models on the market.
>
> -----
> [1] Greenblatt, Ryan, et al. "AI control: Improving safety despite intentional subversion." arXiv preprint arXiv:2312.06942 (2023).
>
> [2] Mathew, Yohan, et al. "Hidden in plain text: Emergence & mitigation of steganographic collusion in LLMs." arXiv preprint arXiv:2410.03768 (2024).
>
> [3] Zolkowski, Artur, et al. "Early Signs of Steganographic Capabilities in Frontier LLMs." arXiv preprint arXiv:2507.02737 (2025).
>
> [4] Kutasov, Jonathan, et al. "SHADE-Arena: Evaluating Sabotage and Monitoring in LLM Agents." arXiv preprint arXiv:2506.15740 (2025).
>
> [5] https://www.lesswrong.com/posts/DCQ8GfzCqoBzgziew/comparing-risk-from-internally-deployed-ai-to-insider-and

---

### Official Review · Reviewer_97WH · 2025-11-01

**Soundness:** 3
**Presentation:** 3
**Contribution:** 3
**Rating:** 6
**Confidence:** 3

**Summary:**

This paper introduces Control Tax, a framework for quantifying the operational and financial costs (“tax”) associated with adding oversight mechanisms—monitors—to AI systems. Using a code backdoor detection setup on the APPS dataset, the authors simulate attacker–monitor dynamics between language models (e.g., GPT-4o, Gemini, o3-mini) and quantify trade-offs between safety, auditing budgets, and cost. They propose a game-theoretic analysis translating ROC curves of monitor performance into long-term safety probabilities, comparing batch and sequential auditing strategies.

**Strengths:**

The introduction of “Control Tax” formalizes an important but underexplored cost-safety trade-off in AI control, bridging empirical LM monitoring with theoretical economics of oversight.

**Weaknesses:**

1. Limited scope of evaluation: Experiments rely solely on the APPS dataset and code tasks, which may not generalize to broader or more agentic AI behaviors.
2. Transferability not discussed. For example, if adaptive attackers can learn the moderator's auditing policy, does this affect safety guarantees?

**Questions:**

1. How sensitive are the conclusions (e.g., optimal auditing strategies) to the specific backdoor definition or dataset used—would agentic or multimodal tasks change the control tax curves?
2. Does the framework extend to adaptive or deceptive attackers that learn the monitor’s auditing policy over time, and how might this affect safety guarantees?

---

> ### Author Response · Authors · 2025-11-26
> **Response to Reviewer 97WH (1/2)**
>
> We are thankful that the Reviewer finds our work to be relevant for “bridging empirical LM monitoring with theoretical economics of oversight.” Below we address the concerns raised by the Reviewer as well as answer the Reviewer’s questions.
> ## Limited scope of evaluation
> Since limited scope of evaluation was a shared concern among reviewers, we have now added an additional evaluation of the Control Tax on another domain. We provide the results and discussion in the newly added Appendix K and Figure 18 (please note  that the appendix is now available not only in the supplementary materials archive but also in the main PDF). For reviewers’ convenience, below we reproduce the results as a table, alongside the adapted discussion from the Appendix.
>
> In the table, the Pareto-optimal monitors are highlighted in bold. Safety is provided with 95% bootstrapped confidence intervals. The attacker model is Gemini 2.5 Pro (Preview).
>
> | Monitor | Avg. cost | Safety |
> | --- | --- | --- |
> | **Gemini 2.0 Flash** | **$5.3** | **0.401 ([0.280, 0.568])** |
> | **Gemini 2.5 Flash Lite** | **$5.5** | **0.417 ([0.324, 0.540])** |
> | GPT 4o Mini | $6.7 | 0.413 ([0.306, 0.548]) |
> | GPT 4.1 Mini | $18.8 | 0.350 ([0.090, 0.443]) |
> | **GPT 5 Mini** | **$20.1** | **0.732 ([0.604, 0.805])** |
> | Gemini 2.5 Flash | $27.0 | 0.655 ([0.558, 0.770]) |
> | o3-mini | $55.3 | 0.334 ([0.244, 0.495]) |
> | GPT 4.1 | $97.1 | 0.562 ([0.422, 0.657]) |
> | GPT 4o | $111.3 | 0.422 ([0.259, 0.572]) |
> | **GPT 5** | **$119.4** | **0.781 ([0.694, 0.867])** |
>
>
> To see how the Control Tax curve would look like beyond the APPS domain, we have adapt our methodology to the monitors from the SHADE-Arena control environment (Kutasov et al [1]). This environment consists of $17$ pairs of long-horizon agentic tasks in detailed simulated environments. Each pair is hand-crafted and consists of a main task and a sabotage task which the agent may or may not be trying to achieve. The tasks are similar to those performed by an administrative assistant, e.g. booking flights or planning a dinner. The sabotage tasks are subtle and context-dependent, e.g. making sure that the flights are booked with a specific airline company despite the company offering highly suboptimal prices.
>
> Re-running the agents on the original tasks is expensive, since one rollout might constitute several hundred tool calls by the agent. However, we can reuse the rollouts generated by Kutasov et al. We took $185$ rollouts ($93$ with a side task and $92$ without) generated with `Gemini 2.5 Pro (Preview)`. We then run a set of models as monitors discriminating the benign trajectories from the ones with the side task. We use the monitoring prompt template from Kutasov et al. Since this monitoring task is more involved than the APPS backdooring task used in the main paper, we also add the most recent generations of closed-source models like `GPT 5`. Finally, we translate the ROC curves of the monitors into safety values at an auditing budget $c=0.05$ using the sequential not-anytime monitoring framework from Section 3.2 in the main paper.
>
> The results are presented in the table above. The main conclusion from the results on APPS persists on SHADE-Arena: for a marginal increase in safety (from $\sim 0.73$ when using `GPT 5 Mini` to $\sim 0.78$ when using `GPT 5`), one needs to pay a large tax (from USD 20 to USD 120). While the high-level takeaway is similar, individual models sometimes behave differently compared to the APPS setting. For example, `o3-mini` does not realize the safety benefits that it brought in APPS. We attribute this to a difference in domains. In APPS, the programs to be evaluated are short, but a well-hidden backdoor might require extensive reasoning to discover. Here, extended reasoning abilities of `o3-mini` play a key role. In SHADE-Arena, on the other hand, monitoring resembles a “needle-in-haystack” problem: discovering sabotage requires going through an extensive transcript of the agent to discover the malicious actions like transferring money to a wrong account. For this, extended reasoning is less important, and instead long-horizon comprehension abilities of newer model families like `Gemini 2.5` and `GPT 5` are more relevant, which makes them perform better in this domain.
>
> ## Adaptive attacks
> While we agree with the Reviewer that adaptive attacks on control protocols more broadly is an important direction for research in AI control, we would like to stress that the strategy of “learning the moderator’s policy” over the course of deployment cannot be applied directly, since one of the assumptions behind AI control is that the attacker does not maintain state between the tasks. This assumption is justified by the fact that the blue team controls the context and the training data of the untrusted model, which the red team simulates in the AI control game. We also discuss this issue in more detail in our response to Reviewer WjCo.

---

> ### Author Response · Authors · 2025-11-26
> **Response to Reviewer 97WH (2/2)**
>
> ## Questions
> > How sensitive are the conclusions (e.g., optimal auditing strategies) to the specific backdoor definition or dataset used—would agentic or multimodal tasks change the control tax curves?
>
> The shape of the optimal strategies mostly depends on the shape of the ROC curve of the monitor and the probability $p_+$ of the attacker succeeding when attempting sabotage. These shapes will be very domain-dependent, but the Control Tax framework will be applicable regardless of the exact shape. The core conclusion that one should audit more heavily at the start has to remain because we have proven it mathematically in Appendix E.
>
> We also discovered some other elements of the ROC-auditing budget empirically. For example, the optimal strategy has discontinuous jumps around the values $p(\tau)$ corresponding to the discontinuities of the _derivative_ of the ROC curve. Concave ROC curves may also lead to batch auditing with a naive strategy underperforming compared to sequential auditing, a phenomenon we discuss in more detail in Appendix C.9.
>
> > Does the framework extend to adaptive or deceptive attackers that learn the monitor’s auditing policy over time, and how might this affect safety guarantees?
>
> The framework of AI control fundamentally relies on the assumption that the attacker is stateless, since the blue team has control over the context of the untrusted model. We kindly refer the Reviewer to our response to Reviewer WjCo, where we discuss this issue in more detail.
>
> -----
> [1] Kutasov, Jonathan, et al. "SHADE-Arena: Evaluating Sabotage and Monitoring in LLM Agents." arXiv preprint arXiv:2506.15740 (2025).

---

### Author Response · Authors · 2025-11-26
**Brief summary of the responses to reviewers**

Dear Reviewers and Area Chair,

We thank the reviewers for the detailed commentary about the paper. We have provided the individual responses to each of the Reviewers. Below, we summarize the raised concerns with our responses for your convenience.

- **Scope of Evaluation (97WH, WjCo, S345)**: We agree with the reviewers that an extra evaluation domain would strengthen the contribution of the paper. We have therefore added evaluation on the SHADE-Arena dataset to Appendix K. The overall conclusion that marginal improvements in safety at the top come at a large cost premium continues to hold in the new domain.
- **Attacker adaptation to the monitor (97WH, WjCo)**: Statelessness of the attacker is a core assumption in AI control, justified by the blue team having control over the untrusted model’s context. Certain subtle violations of it may be present in real deployments but would be out of scope for the current work.
- **Sensitivity to cost (WjCo)**: For internal AI lab deployments, we cannot know the actual costs of running the monitors. We have to therefore settle on a noisy measurement provided by the API costs for external access. Our main contribution, however, is not the exact cost measurement but rather the conceptual framework to be applied by practitioners with their specific costs in mind.
- **Fixed auditing budgets (S345)**: We provide a preliminary discussion of the “sequential anytime” trusted monitoring in Appendix G, where we show how the framework changes if the deployment continues indefinitely (no knowledge about the horizon).
- **Heavy mathematical notation (S345)**: We try to summarize the takeaways from the theoretical discussion in Section 3 at a higher level in Section 4.

To provide further evidence for the stability of our results, the new version of the paper also contains bootstrapped 95% confidence intervals for safety values in Figures 1 and 5.

Unfortunately, only Reviewer S345 had had the opportunity to engage with our responses before the discussion period closed. We would like to note that Reviewer S345 has updated their score from 6 to 8 as a result of our improvements in the paper.

---

### Meta-Review · Area_Chair_WBoU · 2026-01-09

**Summary:**

This paper proposes the notion of Control tax, which could be recognized as the operational and financial cost of integrating control measures into AI pipelines.  This manuscript receives consistently positive reviews. All reviewers acknowledge the contributions, and their reviews support accepting this manuscript (min: 6, max: 6). During the rebuttal period, the authors have carefully addressed all issues raised by different reviewers. And the Reviewer S345 clearly agrees to update the rating. So this paper should be accepted without doubt.

**Reviewer Concerns:**

All concerns from different reviewers have been addressed.

**Reviewer Scores:**

Reviewer S345 clearly agrees to update the rating. And other reviewers provide positive initial ratings.

---

### Decision · Program_Chairs · 2026-01-26

Accept (Poster)